# In Situ Growth of PbS/PbI_2_ Heterojunction and Its Photoelectric Properties

**DOI:** 10.3390/nano12040681

**Published:** 2022-02-18

**Authors:** Shangxun Yang, Jun Han, Jin Zhang, Yingxiu Kong, Huan Liu

**Affiliations:** 1School of Weapons Science and Technology, Xi’an Technological University, Xi’an 710032, China; ysx070619@163.com; 2School of Optoelectronic Engineering, Xi’an Technological University, Xi’an 710032, China; j.zhang@xatu.edu.cn (J.Z.); kongyingxiu@xatu.edu.cn (Y.K.); liuhuan@xatu.edu.cn (H.L.)

**Keywords:** in situ growth, heterojunction, photoresponse, specific detectivity, wide-spectrum detector

## Abstract

In this paper, PbI_2_ thin films with a uniform surface morphology and compact structure were prepared by adjusting the spin coating process parameters. On such a basis, the PbS/PbI_2_ heterojunction was fabricated on the PbI_2_ surface by the method of in situ chemical replacement growth. The results show that the PbS/PbI_2_ heterojunction grown by this method has a clear interface and is closely combined. The introduction of a PbS layer enables its spectral response range to cover the visible and near-infrared regions. Compared with the PbI_2_ thin film device, its responsivity is increased by three orders of magnitude, its response time reduced by 42%, and its recovery time decreased by nearly 1/2 under 450 nm illumination. In the case that there is no response for the PbI_2_ thin film device under 980 nm illumination, the specific detectivity of the PbS/PbI_2_ heterojunction device still amounts to 1.8 × 10^8^ Jones. This indicates that the in situ chemical replacement is a technique that can construct a high-quality heterojunction in a simple process. PbS/PbI_2_ heterojunction fabricated by this method has a visible–near-infrared light detection response range, which provides a new idea for creating visible–near-infrared common-path detection systems.

## 1. Introduction

As an important part of photoelectric information collection, photoelectric detectors are widely used in vehicle driving, motion sensing, security alarms, infrared induction imaging and other aspects [1,2,3,4,5,6,7]. Photoelectric detection systems working within different wavelength ranges have their own advantages in terms of their capability and application. However, with the continuous development of science and technology as well as the emergence of various unforeseen circumstances and interference technologies, users’ requirements for photodetectors are growing. Therefore, photodetectors that only respond in a single-wave band are far from meeting their actual needs. Efficiently obtaining accurate multi-band information in complex environments is extremely important. Thus, the photoelectric detection system with dual-band or multi-band fusion were invented [8,9,10,11,12]. However, the previously developed dual-band or multi-band detection systems are usually composed of two or more independent detection systems. Such systems are costly in terms of production and large in volume, which runs counter to the idea of high integration, miniaturization and lightweight required for detection systems. Therefore, collecting dual-band or multi-band information with a single detector to achieve broad-spectrum common-path detection is the key to solving the above problems [13,14,15].

In order to expand the response range of photoelectric devices and improve their responsivity, a good few scientists have proposed many schemes of referential meaning. For example, the heavily doped silicon photoelectric device fabricated by Aurore J. Said et al. achieved an external quantum efficiency of 3000% at 12 V bias voltage, with a detection range from the visible region to the near-infrared region [16]. Although such a method of doping traditional silicon substrate and preparing a microstructure can widen the response range of the device, the preparation method is complex and will bring about a substantial number of defects in the silicon material, thus amplifying the dark current of the photoelectric device and decreasing its responsivity and sensitivity. Therefore, there exist some limits for the photoelectric device fabricated with a single material. Wang et al. doped PTB7-Th in the P3HT:PC71BM system and successfully obtained a highly sensitive polymer photoelectric device with a spectral response range from the ultraviolet region to the near-infrared region [17]. Photoelectric devices fabricated with organic semiconductors have an ultra-wide detection spectral range, but the instability of organic materials will make their service life much shorter than those fabricated with inorganic materials. Zhuo et al. prepared GaN thin films on sapphire substrates in the method of MOCVD, and stacked GaN with β-In_2_Se_3_ obtained by mechanical exfoliation to fabricate the photodetector with a heterojunction structure [18]. It achieved responses in the UV and near-infrared regions with a responsivity of 1.6 A/W under 365 nm and that of 0.03 A/W under 850 nm as well. The process of fabricating heterojunction is complex. The size of the device is limited by that of the tube furnace chamber, and the cleanliness of the stack surface is highly required in the physical stacking process. Mukherjee et al. realized the in situ growth of high-quality 0D/2D mixed dimensional heterojunction structure in MoS_2_ by the low-temperature hydrothermal method [19]. Therefore, the device prepared with the heterojunction had responses at 400~1600 nm with a specific detectivity of 10^12^ Jones. The process of fabricating heterojunction by this method is simpler than that by the chemical vapor deposition (CVD) method, and the as-prepared suspension can be fabricated with large devices by spin coating, spraying and other methods. However, two-dimensional materials in the suspension are small in size, stacking may arise under certain circumstances, and the agglomeration of quantum dots may occur to some extent. In addition, the preparation of the carrier transport layer is required as an auxiliary for its preparation with large devices.

In this paper, a simpler method for fabricating visible–near-infrared photodetectors was proposed. The heterojunction was constructed through the in situ growth of the PbS layer on the surface of the PbI_2_ thin film. Therein, the two materials were responsible for the absorption of and response in different wavebands, respectively. The spectral response range of the photodetectors was expanded. The compact PbI_2_ thin film also had excellent carrier mobility, which can act as a carrier migration layer. This study provides an idea and method with great application potential for fabricating array photodetectors with all-day and wide-spectrum response.

## 2. Materials and Methods

Figure 1 shows the specific preparation process of the PbI_2_ thin film and PbI_2_/PbS heterojunction. The PbI_2_ solution was prepared at room temperature with N, N-dimethylformamide (DMF) (Alfa Aesar, Shanghai, China) and dimethyl sulfoxide (DMSO) (Alfa Aesar, Shanghai, China) mixed solvent. The prepared solution was magnetically stirred for 12 h at 70 °C and 100 r/min, and was aged for 48 h. Then, the prepared solution was spin-coated on the surface of the cleaned sodium calcium glass at a rotational speed of 6000 r/min for 30 s, and annealed at 80 °C for 20 min to obtain the PbI_2_ thin film. Finally, the PbI_2_ thin film was put into 1 mg/mL of Na_2_S (Alfa Aesar, Shanghai, China) ethanol solution and taken out, and the PbI_2_/PbS heterojunction was obtained after the surface of the sample was cleaned with anhydrous ethanol (Alfa Aesar, Shanghai, China) and dried. The purities of PbI_2_, Na_2_S, DMF, and DMSO are 99, 98, 99.9, and 99%, respectively.

The phase characteristics of samples was tested through X-ray diffraction (XRD) analysis (D8 ADVANCE, Bruker AXS, Karlsruhe, German). The morphological property was observed with a field-emission scanning electron microscopy (GeminiSEM 500, Zeiss, Aalen, Germany). The laser Raman spectroscope (LabRAM HR Evolution, HORIBA, Fukuoka, Japan) was used for Raman testing and analysis with a 532 nm laser for excitation. The absorption spectra were collected by using the UV-NIR spectrometer (JASCO V-570 UV/vis/NIR, JASCO, Tokyo, Japan). Spectral response characteristics of samples were measured with optical spectrum analyzer (DSR-F4-XIAN, Zolix, Beijing, China). The I–V curves and time-resolved photoresponse of the samples were obtained with a self-organized detection system which includes two digital source tables (2280 and 2450 Keithley, Cleveland, OH, USA), and two monochromatic light emitting diodes (450 and 980 nm, Shanghai, China). The optical power meter (843-R, Newport, CA, USA) was used for light intensity calibration.

## 3. Results and Discussion

In this study, PbI_2_ thin film was used as precursor, and a layer of PbS thin film was formed in situ on the surface of the PbI_2_ thin film through the replacement reaction between PbI_2_ and Na_2_S, thus forming the PbS/PbI_2_ heterojunction. As shown in Figure 1, before and after the replacement reaction, the surface color of the sample changed from light yellow to dark brown due to the difference in absorption spectra between PbS and PbI_2_.

In order to obtain a PbS/PbI_2_ heterojunction with excellent performance, the quality of the PbI_2_ thin film as a precursor is essential. In this paper, the effects of PbI_2_ concentration and the ratio of dimethyl sulfoxide (DMSO) to dimethyl formamide (DMF) on the quality of PbI_2_ thin film in its preparation were systematically studied. As shown in Appendix A, a transparent, clear and precipitate-free PbI_2_ solution can only be obtained when the ratio of DMF to DMSO is 9:1 after aging for two days. In the solvent with such a ratio, PbI_2_ concentration cannot exceed 1.75 M, otherwise precipitation will be produced, as shown in Appendix A. In accordance with Appendix A, as the concentration of PbI_2_ solution increases from 1 M to 1.75 M, PbI_2_ thin film becomes more compact and smoother, and its thickness rises from approximately 70 to approximately 300 nm. The visible–near-infrared spectrophotometer test shows that the PbI_2_ thin film prepared with 1.75 M of PbI_2_ solution has the best visible light absorption, as shown in Appendix A.

In an effort to investigate the effects of the in situ growing time of PbS on the photon absorbing capacity of the PbS/PbI_2_ heterojunction in the near-infrared band, the growing time was set to 5, 10, 15, 20, 25, and 30 s, respectively. As shown in Appendix A, the PbS/PbI_2_ heterojunction has the best absorption in the near-infrared band when the growing time was 20 s. Furthermore, if the growing time exceeds 30 s, some part of the PbI_2_ thin film substrates will be dissolved and exfoliated by anhydrous ethanol, as shown in Appendix A.

In view of the above results, the process parameters of preparing PbI_2_ thin film and PbS/PbI_2_ heterojunction were determined in the paper as follows: the PbI_2_ precursor thin film was prepared with the solvent of which the ratio of DMF to DMSO was 9:1 and PbI_2_ solution with a concentration of 1.75 M; PbI_2_ precursor thin film was soaked in Na_2_S ethanol solution for 20 s. The SEM images of PbI_2_ thin film and PbS/PbI_2_ heterojunction prepared by the above process parameters is shown in Figure 2, they both have smooth and compact surfaces, as well as a clear heterojunction interface with the thicknesses of the PbS layer and PbI_2_ layer being approximately 25 and 230 nm, respectively.

To verify whether the samples were the target products, they were characterized with XRD and Raman spectra. Figure 3a shows the XRD curves of the prepared PbI_2_ thin film and PbS/PbI_2_ heterojunction. Compared with the powder diffraction standard cards of PbI_2_ and PbS, it was found that there exists a one-to-one correspondence between the XRD diffraction peak of the PbI_2_ thin film and the standard cards of PbI_2_. A very weak diffraction peak appears at 30° in the XRD spectra of the PbS/PbI_2_ heterojunction. In comparison with the standard card of PbS, this diffraction peak should correspond to the (200) crystalline of PbS. Since the diffraction peak of PbS is very weak, it is impossible to completely determine whether the PbS layer has been deposited on the surface of the sample. Therefore, the samples are also characterized with Raman spectra. Figure 3b shows the Raman spectra curve of the as-prepared PbS/PbI_2_ heterojunction, and the inset indicates the comparison of the Raman spectra curves between PbI_2_ thin film and PbS/PbI_2_ heterojunction within the range of 50~300 cm^−1^. According to the curves, the Raman peaks of PbI_2_ are 70.57, 93.97, 109.32, 165.69, and 214.26 cm^−1^, respectively [20,21,22]. Through comparison, it was found that the Raman peaks of PbS/PbI_2_ heterojunction also appear at 70, 93, and 109 cm^−1^, and the vibration peak at 1377.55 cm^−1^ is a unique Raman peak of PbS [23,24]. Based on the characterization and analysis of XRD and Raman, it can be concluded that the PbS layer is indeed generated on the surface of PbI_2_, hence forming PbS/PbI_2_ heterojunction.

Figure 4a shows the visible–near-infrared absorption spectra curves of the PbI_2_ thin film and PbS/PbI_2_ heterojunction. In accordance with the figure, the optical absorption of the PbS/PbI_2_ heterojunction in the visible and near-infrared regions was significantly improved compared with that of PbI_2_ thin film. The insets in Figure 4a are the band gap width of the PbI_2_ thin film and PbS/PbI_2_ heterojunction fitted by Tauc plot method [25,26,27]. PbI_2_ and PbS/PbI_2_ are the direct band gaps with widths of 2.41 and 0.7 eV, respectively. The measured band gap of the PbI_2_ thin film is close to the theoretical value (2.32 eV). However, the band gap of PbS/PbI_2_ is mainly determined by the upper PbS, so the fitting results deviate from the theoretical band gap of PbS (0.37 eV). There are two possible reasons: firstly, PbS quantum dots which are formed through the in situ replacement reaction have a quantum confinement effect, leading to an increase in the band gap; secondly, because the PbS layer is very thin, the measured absorption spectra curve is the result of the combined action of PbS and PbI_2_, resulting in a larger fitting band gap. Figure 4b is the normalized photoresponse curve of PbS/PbI_2_ heterojunction at 30 V external bias. It can be obviously observed that PbS/PbI_2_ heterojunction has a response in the visible and near-infrared region, and the responsivity within the range of 400~515 nm is obviously higher than that in the range after 515 nm. The main reason for that is PbS and PbI_2_ can jointly generate the photocurrent within the range of 400~515 nm, while only PbS creates the photocurrent after 515 nm (2.41 eV, the energy gap of PbI_2_, corresponds to the absorption cutoff wavelength of 515 nm).

In order to further study the characteristics of the visible–near–infrared photoresponse of the PbI_2_ thin film and PbS/PbI_2_ heterojunction devices, an interdigital electrode was etched with a channel width of 0.20 mm on FTO glass, and the PbI_2_ thin film and PbS/PbI_2_ heterojunction were prepared on it, as shown in Figure 5a. Figure 5b shows the I–V curves of the PbS/PbI_2_ heterojunction device under 450 and 980 nm illumination. It can be obviously observed that the PbS/PbI_2_ heterojunction device has obvious characteristics of photosensitive resistance, and the photocurrent under 450 nm is clearly greater than that under 980 nm. The reason is that the photocurrent under 450 nm is jointly provided by PbI_2_ and PbS, while the photocurrent under 980 nm is only provided by PbS, which is consistent with the results shown in Figure 4b. Moreover, the photocurrent of the PbS/PbI_2_ heterojunction device is amplified with the increase in external bias voltage, which is a typical characteristic of photosensitive devices. However, when the voltage exceeds 30 V, the device will break down. Therefore, the external bias voltage applied in this study was determined as 30 V.

Under the bias voltage of 30 V, the PbI_2_ thin film and PbS/PbI_2_ heterojunction devices were periodically irradiated by monochromatic light at 450 and 980 nm wavelengths, respectively. The obtained dynamic time-resolved optical response curves are shown in Figure 6a,b. By comparison, it can be found that both devices have responses under 450 nm illumination. It is worth noting that the response current of PbS/PbI_2_ heterojunction is nearly three orders of magnitude higher than that of the PbI_2_ thin film. However, the PbI_2_ thin film device fails to respond under 980 nm illumination, while the PbS/PbI_2_ heterojunction device still has a photocurrent of 10 μA. This shows that the fabricated PbS/PbI_2_ heterojunction device couples the photoelectric properties of the two materials, and retains the photoresponse capabilities of PbI_2_ and PbS in the visible and near-infrared bands, respectively. As shown in Figure 6c, the response and recovery times of the PbI_2_ thin film device are 0.69 s and 0.80 s, respectively, under 450 nm illumination. Under the same conditions, those of the PbS/PbI_2_ heterojunction device are 0.40 s and 0.42 s, respectively. The response time is reduced by 42%, and the recovery time nearly by 1/2. The reason for that is because the heterojunction can better separate photogenerated carriers and accelerate the hole and electron transport and collection. The response and recovery times of the PbS/PbI_2_ heterojunction device are 0.25 and 0.45 s, respectively, under 980 nm illumination, as shown in Figure 6d. Compared with that under 450 nm, the response time of the device is shorter, which indicates that the PbI_2_ layer has excellent carrier transport properties.

The responsivities of the above two devices were calculated by using the formula:*R* = *I_light_*/*P_in_*(1)

(*I_light_* refers to the photogenerated current, and *P_in_* the incident light power). Then, the specific detectivity of the device was calculated through formula:(2)$$D^{*}=\frac{R\sqrt{S}}{\sqrt{2qI_{dark}}}$$
where *R* refers to the responsivity, *S* the effective light area, *q* the charge constant, and *I_dark_* the dark current [28]. Figure 7 shows the responsivity and specific detectivity of the PbI_2_ thin film and the PbS/PbI_2_ heterojunction devices under incident light at different wavelengths and power densities. According to Figure 7a,b, the responsivities of the two devices increase with the decrease in light power density under 450 nm illumination, indicating that both devices have excellent characteristics of low-illumination response. The main reason is because at a low light power density, the defects of the material will capture some charges to produce a local electric field, leading to the photogating effect. This makes carriers operate in the channel and complicates the chances of recombining them, so that the carriers’ lifetime is prolonged, resulting in high gain under low light. The combination rate of the carriers increases under high power illumination, and the defects in the material are filled by carriers excited by strong light resulting in a decrease in responsivity [29,30,31]. Particularly, when the light power density is as low as 0.00667 mw/cm^2^, the responsivity and specific detectivity of the PbI_2_ thin film device are 2400 μA/W and 1.8 × 10^10^ Jones, respectively, while those of the PbS/PbI_2_ heterojunction device are 740 mA/W and 1.4 × 10^11^ Jones, respectively. Compared with the PbI_2_ thin film device, the responsivity of the PbS/PbI_2_ heterojunction device improves by more than 300 times and the specific detectivity by over 8 times.

Figure 7c,d show the responsivity and specific detectivity of the PbS/PbI_2_ heterojunction device at different light power densities under 980 nm illumination. Values of its responsivity and specific detectivity rise first and then fall as the light power density grows. When the light power density is 43.92 mw/cm^2^, the responsivity and specific detectivity are 1.36 mA/W and 1.8 × 10^8^ Jones, respectively. In summary, the PbS/PbI_2_ heterojunction device has low illumination response characteristics in the visible region. Moreover, its photoresponse range extends to the near-infrared region. This device has a full-day-wide spectral detection capability.

## 4. Conclusions

In this study, the PbS/PbI_2_ heterojunction structure was successfully prepared by the method of growing PbS on a PbI_2_ thin film by in situ chemical substitution reaction. Compared with the PbI_2_ thin film, the spectral response range of the photoelectric device with the PbS/PbI_2_ heterojunction structure expanded from the previous visible light band to the visible–near-infrared band; the responsivity of the device in the visible light band rose from 2.4 × 10^3^ μA/W to 7.4 × 10^2^ mA/W, with an increase of two orders of magnitudes; the specific detectivity improved from 1.8 × 10^10^ Jones to 1.4 × 10^11^ Jones, an increase of one order of magnitude; and the response/recovery time decreased by 45% from 1.49 to 0.82 s. Moreover, the response/recovery time, responsivity, and specific detectivity of the PbS/PbI_2_ heterojunction device in the near-infrared band also reached 0.7 s, 1.09 mA/W, and 1.8 × 10^8^ Jones, respectively. The PbS/PbI_2_ heterojunction device has noticeably low light response characteristics, especially in the visible region. These strengths preview the great potential of PbS/PbI_2_ heterojunction devices in the development of visible–near-infrared all-day spectral photoelectric devices.

## Figures and Tables

**Figure 1 nanomaterials-12-00681-f001:**
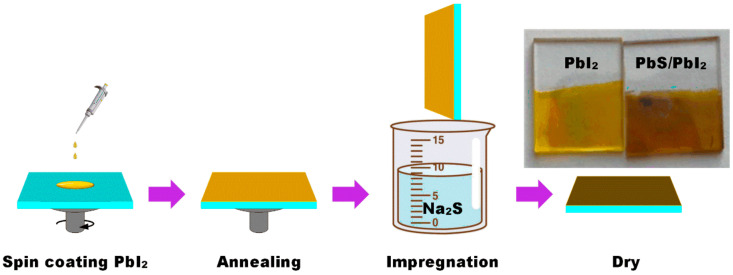
Schematic of preparing the PbS/PbI_2_ heterojunction.

**Figure 2 nanomaterials-12-00681-f002:**
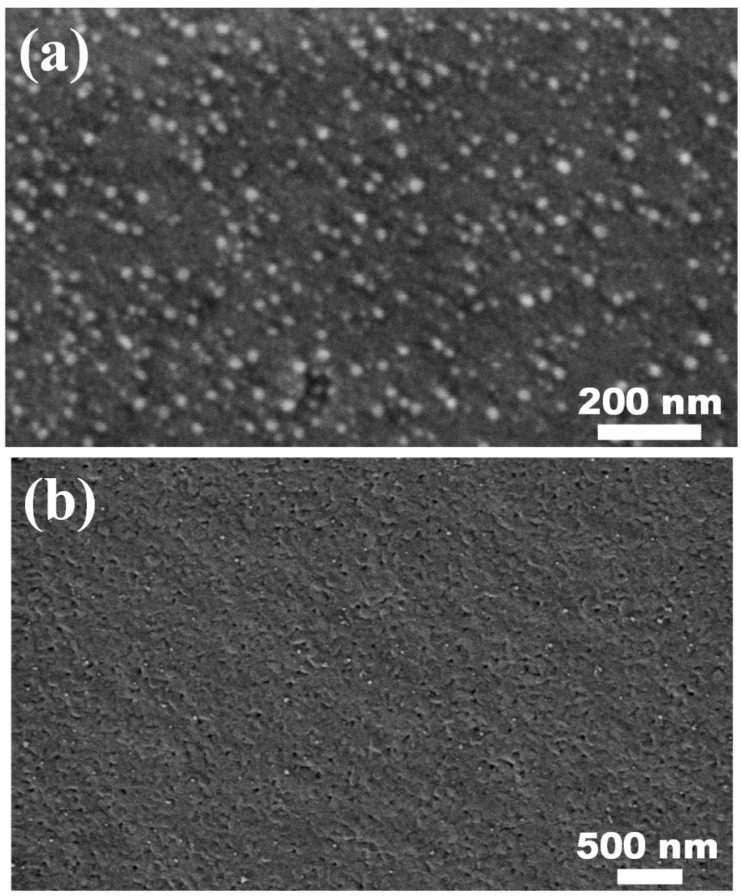
SEM patterns of the PbI_2_ thin film and PbS/PbI_2_ heterojunction: (**a**) top view of the PbI_2_ thin film; (**b**) top view of the PbS/PbI_2_ heterojunction; and (**c**) cross-section of the PbS/PbI_2_ heterojunction.

**Figure 3 nanomaterials-12-00681-f003:**
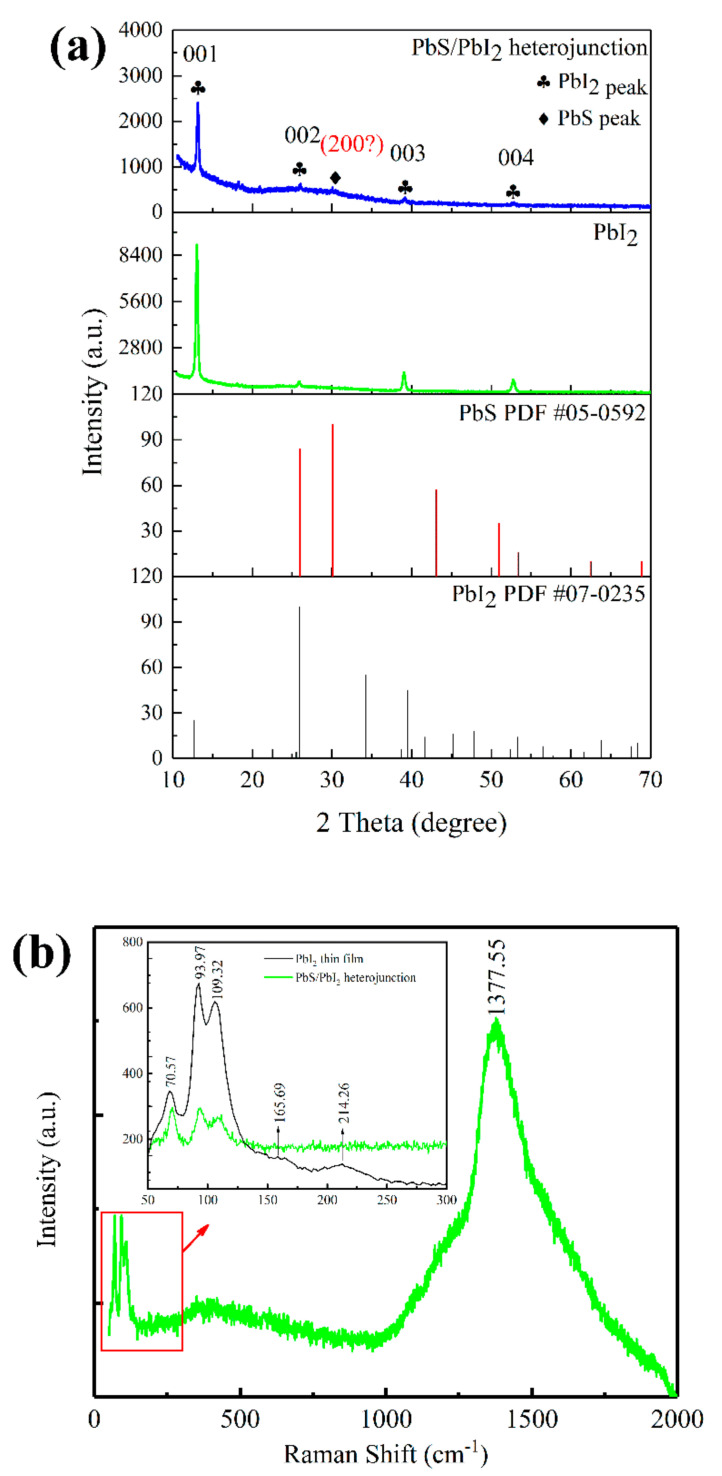
XRD and Raman curves of PbI_2_ thin film and PbS/PbI_2_ heterojunction: (**a**) XRD curves; (**b**) Raman curves and the inset is the comparison of Raman spectra between PbI_2_ thin film and PbS/PbI_2_ heterojunction within the range of 50–300 cm^−1^.

**Figure 4 nanomaterials-12-00681-f004:**
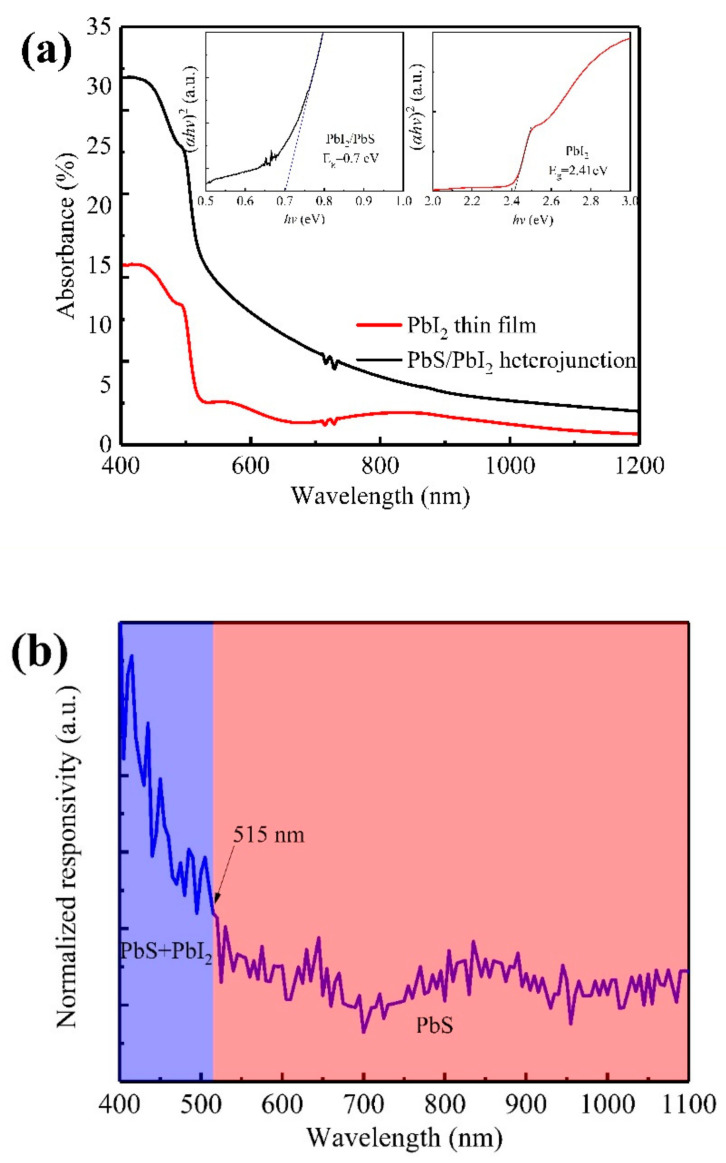
(**a**) Vis–NIR absorption spectra of the PbI_2_ thin film and PbS/PbI_2_ heterojunction, and the insets are the band gap fitting curves of the PbI_2_ thin film and PbS/PbI_2_ heterojunction; and (**b**) normalized responsivity of the PbS/PbI_2_ heterojunction.

**Figure 5 nanomaterials-12-00681-f005:**
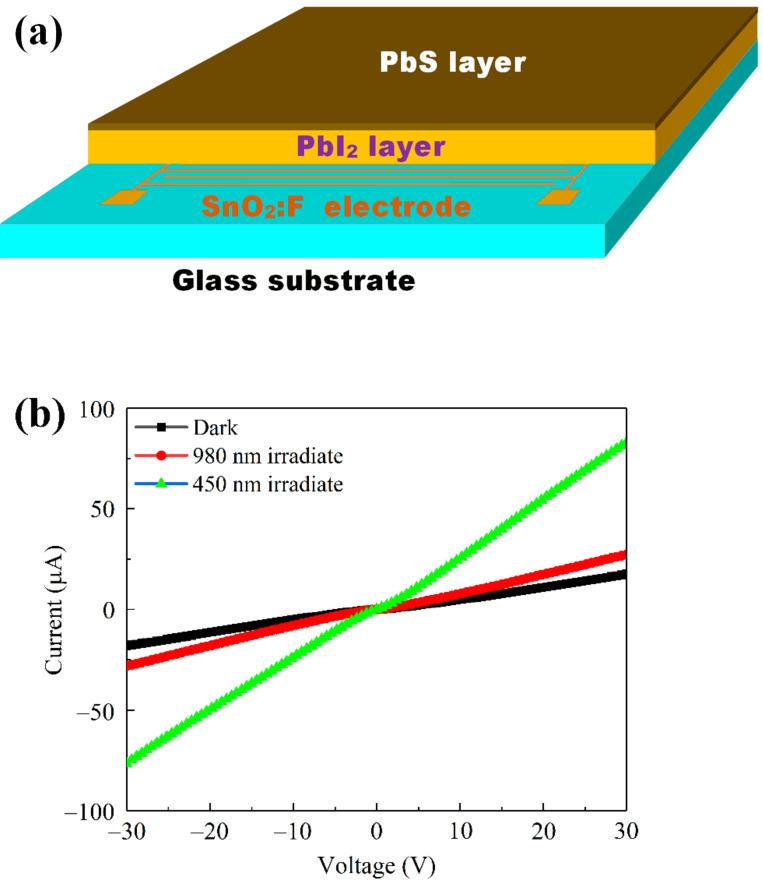
(**a**) Schematic of the PbS/PbI_2_ heterojunction device structure; and (**b**) I−V curves of the PbS/PbI_2_ heterojunction device under 450 and 980 nm incident light.

**Figure 6 nanomaterials-12-00681-f006:**
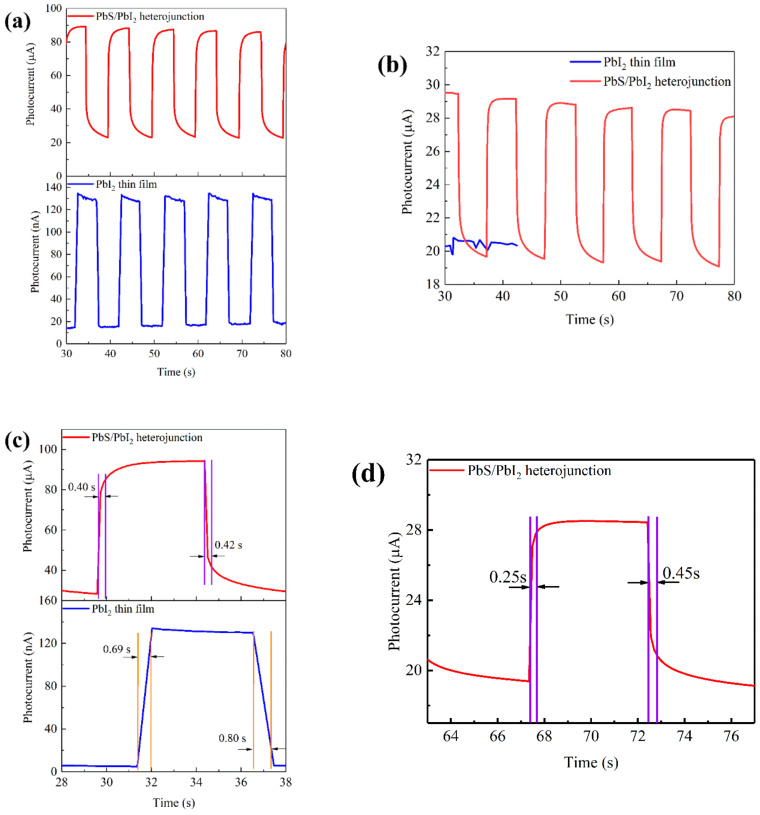
Dynamic time-resolved photoresponse and response/recovery plots of the PbI_2_ thin film device and the PbS/PbI_2_ heterojunction device under periodical illumination by monochromatic light source at different wavelengths and with 30 V bias voltage: (**a**) dynamic time-resolved photoresponse under 450 nm illumination; (**b**) dynamic time-resolved photoresponse under 980 nm illumination; (**c**) response/recovery plots under 450 nm illumination; and (**d**) response/recovery plot under 980 nm illumination.

**Figure 7 nanomaterials-12-00681-f007:**
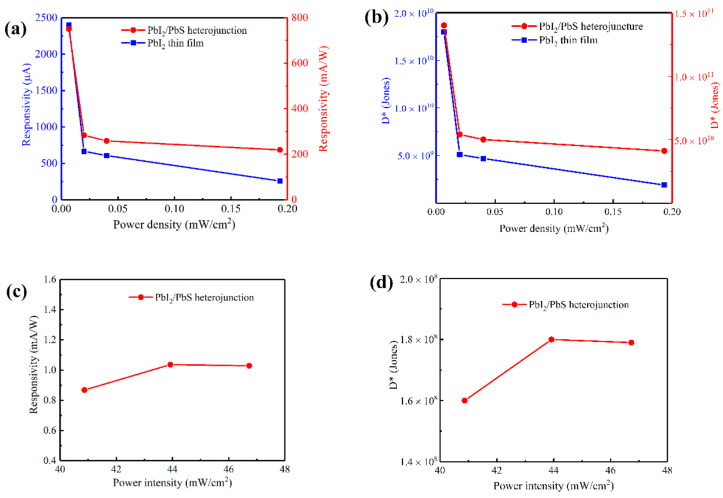
Responsivity and specific detectivity of the PbI_2_ thin film device and the PbS/PbI_2_ heterojunction device under illumination at different wavelengths and the power density of incident light at 30 V bias voltage: (**a**) relationship between responsivity and power density under 450 nm illumination; (**b**) relationship between specific detectivity and power density under 450 nm illumination; (**c**) relationship between responsivity and power density under 980 nm illumination; and (**d**) relationship between specific detectivity and power density under 980 nm illumination.

## Data Availability

Not applicable.

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
