# Peer review of "In Situ Growth of PbS/PbI2 Heterojunction and Its Photoelectric Properties"

_nanomaterials, 2022, doi:10.3390/nano12040681_

Round 1

Reviewer 1 Report

Figure 3 (a) shows the XRD of the heterojunction. First, the figure does not show arbitrary units but no units. Second, more concerning is the claim of the 200 PbS peak. It's not really seen (only with wishful thinking) in the broad hump from about 20 to 35 degree. If it is really there, PbS - more than likely - is amorphous. The authors did not comment on the weak XRD response of PbS. More elaboration about the PbS quality is required before the paper can be published.   

Author Response

We fully agree with the reviewer's point of view. The very weak diffraction peak near 30° in the XRD curve of PbS/PbI2 heterojunction may be correspond to the (200) crystalline of PbS. Since the diffraction peak of PbS is very weak, it is impossible to completely determine whether the PbS layer has been deposited on the surface of the sample. Therefore, the samples are also characterized with Raman spectra. Raman curves are known as the "fingerprint" information of materials, which can accurately characterize the type of material. Based on the characterization and analysis of Raman, it can be concluded that PbS layer is indeed generated on the surface of PbI2, hence forming PbS/PbI2 heterojunction. According to the comments of reviewers, we have modified the XRD pattern and added relevant instructions in the revised manuscript.

Reviewer 2 Report

The manuscript requires revision as there are changes to be made. The following issues must be addressed: 1. Introduction part should be significantly improved with more references and discussions about similar or complementary papers. Examples of other heterostructures obtained by different methods can be useful (e.g. DOI: 10.1016/j.apcatb.2014.06.059, DOI: 10.1016/j.cplett.2020.137926, DOI: 10.1016/j.jpcs.2019.06.008). 2. The authors should explain each abbreviation (DMF, DMSO). The purity of each substance should be provided. 3. Both figs 2 a and b are not clear. Please provide higher resolution SEM images. 4. I recommend EDS or XPS analysis to confirm the existence of PbS at the PbI2 surface. 5. What about photocurrent generation without external bis? 6. Mott-Schottky and impedance measurements will be useful to understand the electrical behavior of this sample. 7. Conclusion part should be more comprehensive. The most relevant results should be included.

Author Response

The manuscript requires revision as there are changes to be made. The following issues must be addressed:

  1. Introduction part should be significantly improved with more references and discussions about similar or complementary papers. Examples of other heterostructures obtained by different methods can be useful (e.g. DOI: 10.1016/j.apcatb.2014.06.059, DOI: 10.1016/j.cplett.2020.137926, DOI: 10.1016/j.jpcs.2019.06.008).

We are indeed very much grateful to the reviewer for pointing out this issue. According to the reviewer’s request, the examples of other heterostructures obtained by different methods have been presented in “Introduction”, and necessary references have also been added.

  1. The authors should explain each abbreviation (DMF, DMSO). The purity of each substance should be provided.

We apology for our negligence in the original manuscript. The full name of DMF and DMSO, and purity of each substance have been added in the revised manuscript.

  1. Both Figs 2 a and b are not clear. Please provide higher resolution SEM images.

The higher resolution SEM images have been offered in the Fig. 2 (a) and (b).

  1. I recommend EDS or XPS analysis to confirm the existence of PbS at the PbI2 surface.

An attempt was made to adopt EDS mapping to qualitatively and quantitatively analyze the types and contents of elements in PbI2/PbS heterojunction, but it led nowhere. There are two reasons. Firstly, the mapping function of the scanning electron microscope requires a minimum thickness of 500 nm for the substrate, while the overall thickness of the heterojunction prepared by us is about 300 nm, which cannot be detected. Secondly, in the mapping mode, the respective scanning peaks of Pb and S are extremely similar and almost overlapped, making it difficult to identify the scanned elements. Therefore, we failed to obtain the analysis results of EDS mapping. In terms of qualitative analysis, Raman curves are known as the "fingerprint" information of materials, which can accurately characterize the type of material. The results of Raman spectrogram and XRD can directly indicate that the heterojunction prepared by us is composed of PbI2 and PbS. In SEM images, the interface and thickness of PbI2 layer and PbS layer can be clearly observed due to different reflection of electrons by different materials. Therefore, it is believed that the characterization of material phase in this paper is complete.

  1. What about photocurrent generation without external bis?

In the absence of external bias voltage, the photocurrent is almost not detected. This is because the electrode structure we used is a waveguide electrode, and the fabricated photoelectric device is in the category of photosensitive resistor. The effect of optical input on the device is mainly demonstrated in the resistance of the device, whereas bias voltage amplifies this effect. Therefore, the photocurrent cannot be detected without current.

  1. Mott-Schottky and impedance measurements will be useful to understand the electrical behavior of this sample.

We are indeed very much grateful to the reviewer for this suggestion. However, our laboratory does not have the corresponding equipment, and we hope to use these equipment in future research.

  1. Conclusion part should be more comprehensive. The most relevant results should be included.

We are indeed very much grateful to the reviewer for pointing out this issue. According to the reviewer’s request, the “Conclusions” has been rewritten in the revised vision.

Reviewer 3 Report

Yang et al. show the novel strategy for the in-situ growth of PbS/PbI2 heterojunction in conjunction with fast visible near-infrared detection. Overall,  the  proposed  facile  fabrication  strategy  and  superior  device  performance  exhibit  high  potential  for  boosting  the  applications  in  low-dimensional photoelectronic. Even though detectivity is a few orders lower than the reported values of PbS/PbI2 system still has a chance to improve that via post-treatment of heterojunction film.  Authors try to thoroughly characterize the films. Although very good performance is obtained, the works seem to miss some originality and characterizations that can be adopted by other scientists. I, therefore, would like to consider the following points:

  1. Need to try post-treatment to further boost the performance
  2. EDS mapping and XPS will give a more reliable surface characterization of heterojunction structure
  3. Figure 5 (a) Schematic of the PbS/PbI2 heterojunction device structure needs to be redrawn, current drawing seems bilayer structure rather than heterojunction structure.
  4. Author should not claim visible infrared detection in the paper, should write visible near-infrared detection.

Author Response

Yang et al. show the novel strategy for the in-situ growth of PbS/PbI2 heterojunction in conjunction with fast visible near-infrared detection. Overall, the proposed facile fabrication strategy and superior device performance exhibit high potential for boosting the applications in low-dimensional photoelectronic. Even though detectivity is a few orders lower than the reported values of PbS/PbI2 system still have a chance to improve that via post-treatment of heterojunction film. Authors try to thoroughly characterize the films. Although very good performance is obtained, the works seem to miss some originality and characterizations that can be adopted by other scientists. I, therefore, would like to consider the following points:

  1. Need to try post-treatment to further boost the performance

Thank you very much for the suggestions put forward by reviewers, but we find that PbS samples are easy to be oxidized when annealed in atmospheric environment, and our laboratory does not have experimental equipment such as vacuum annealing or glove box for the time being. We also hope to improve the device performance in the following aspects in the future work: the device will be further optimized in such methods as annealing the prepared heterojunction at higher temperature and doping oxygen in the PbS layer so as to enhance the response ability of the device in the near infrared band; or as replacing the semiconductor electrode with graphene electrode or metal electrode to improve the collection efficiency of carriers; or as reducing the channel width of the electrode to achieve that.

  1. EDS mapping and XPS will give a more reliable surface characterization of heterojunction structure

An attempt was made to adopt EDS mapping to qualitatively and quantitatively analyze the types and contents of elements in PbI2/PbS heterojunction, but it led nowhere. There are two reasons. Firstly, the mapping function of the scanning electron microscope requires a minimum thickness of 500 nm for the substrate, while the overall thickness of the heterojunction prepared by us is about 300 nm, which cannot be detected. Secondly, in the mapping mode, the respective scanning peaks of Pb and S are extremely similar and almost overlapped, making it difficult to identify the scanned elements. Therefore, we failed to obtain the analysis results of EDS mapping. In terms of qualitative analysis, Raman curves are known as the "fingerprint" information of materials, which can accurately characterize the type of material. The results of Raman spectrogram and XRD can directly indicate that the heterojunction prepared by us is composed of PbI2 and PbS. In SEM images, the interface and thickness of PbI2 layer and PbS layer can be clearly observed due to different reflection of electrons by different materials. Therefore, it is believed that the characterization of material phase in this paper is complete.

  1. Figure 5 (a) Schematic of the PbS/PbI2 heterojunction device structure needs to be redrawn, current drawing seems bilayer structure rather than heterojunction structure.

The combination of the two materials is more compact than low-dimensional materials which are combined by Van der Waals force. Because waveguide electrodes, of which the electrode structure featuring horizontal transmission is weaker than that featuring longitudinal transmission in junction characteristics, are used instead of junction electrodes with an up-down structure, the junction characteristics are not obviously demonstrated in the I-V curves. To this end, the I-V curves of the photocurrent at 450 nm are extracted, as shown in the figure below, and the part near point 0 is amplified. According to the amplified figure, it can be clearly seen that the junction characteristics of the heterojunction near point 0 are relatively noticeable, which indicates that the structure we prepared is a heterojunction structure rather than a simple stacked structure of two materials.

  1. Author should not claim visible infrared detection in the paper, should write visible near-infrared detection.

We are very sorry for our previous negligence; this error has been corrected in the revised manuscript.

Round 2

Reviewer 2 Report

The manuscript can be published in the present form.

Reviewer 3 Report

Thank you for your revised manuscript. It can be accepted in the present form.